# Impact of Fermentable Fibres on the Colonic Microbiota Metabolism of Dietary Polyphenols Rutin and Quercetin

**DOI:** 10.3390/ijerph16020292

**Published:** 2019-01-21

**Authors:** Bahareh Mansoorian, Emilie Combet, Areej Alkhaldy, Ada L. Garcia, Christine Ann Edwards

**Affiliations:** Human Nutrition, School of Medicine, Dentistry and Nursing, College of Medical Veterinary and Life Sciences, University of Glasgow, Glasgow G31 2ER, UK; Bahareh.mansoorian@gmail.com (B.M.); Areej.alkhaldy@gmail.com (A.A.), Emilie.CombetAspray@glasgow.ac.uk (E.C.); Ada.Garcia@glasgow.ac.uk (A.L.G.)

**Keywords:** fibre, fermentation, microbiome, colon, microbiota, short-chain fatty acids, polyphenols, rutin, quercetin, phenolic acids

## Abstract

Dietary fibre and polyphenols are both metabolised to short-chain fatty acids (SCFAs) and phenolic acids (PA) by the colonic microbiota. These may alter microbiota growth/diversity, but their interaction is not understood. Interactions between rutin and raftiline, ispaghula or pectin were investigated in human faecal batch cultures (healthy participants; 19–33 years, 4 males, 6 females, BMI 18.4–27.4) after a low (poly)phenol diet three days prior to study. Phenolic acids were measured by gas chromatography-mass spectrometry and SCFAs by gas chromatography-flame ionisation after 2, 4, 6, and 24 h. Rutin fermentation produced Phenyl acetic acid (PAA), 4-Hydroxy benzoic acid (4-OHBA), 3-Hydroxy phenyl acetic acid (3-OHPAA), 4-Hydroxy phenyl acetic acid (4-OHPAA), 3,4-Dihydroxy phenyl acetic acid (3,4-diOHPAA), 3-Hydroxy phenyl propionic acid (3-OHPPA), and 4-Hydroxy phenyl propionic acid (4-OHPPA). 3,4-DiOHPAA and 3-OHPAA were predominant at 6 h (1.9 ± 1.8 µg/mL, 2.9 ± 2.5 µg/mL, and 0.05 ± 0.0 µg/mL, respectively) and 24 h (5.5 ± 3.3 µg/mL, 3.1 ± 4.2 µg/mL, and 1.2 ± 1.6 µg/mL). Production of all PA except 3-OHPPA and 4-OHPPA was reduced by at least one fibre. Inhibition of PA was highest for rutin (8-fold, *p* < 0.01), then pectin (5-fold, *p* < 0.01), and ispaghula (2-fold, *p* = 0.03). Neither rutin nor quercetin had a detectable impact on SCFA production. These interactions should be considered when assessing dietary polyphenols and potential health benefits.

## 1. Introduction

The role of the gut microbiome in health may be enacted by the action of the bioactive molecules they release in the human colon. Important substrates for the gut microbiota include plant polyphenols and dietary fibre and the main bioactive molecules released include phenolic acids (PA) and short-chain fatty acids (SCFAs), respectively. Common dietary polyphenols include rutin and quercetin, which are found in fruits and vegetables such as tomatoes, onions, kale, tea, some berries, citrus fruits, and apricots [1,2]. These and other flavonoids have been associated with reduced risk of cardiovascular disease and other health benefits. In a meta-analysis of 18 randomised clinical trials [3], flavonol intake, mainly quercetin, reduced total cholesterol, triacylglyceride (TAG), low-density lipoprotein (LDL), systolic blood pressure (SBP), diastolic blood pressure (DBP) and increased high-density lipoprotein (HDL) levels. A meta-analysis of seven case-control studies showed an inverse association between flavonol and procyanidin intake and development of colorectal cancer [4].

Bioavailability is a key factor influencing the health promoting potential of plant (poly)phenolics. The majority of polyphenols are not absorbed (only 7.2% of ingested polyphenols are found in blood and tissues) [5]. Thus, most ingested polyphenols reach the colon, where they are catabolised to PA by the colonic microbiota [6,7,8,9,10], although inter-individual variations do exist [11]. These catabolites may be responsible for some of the health benefits attributed to polyphenol-rich foods [7,8,12].

Colonic metabolism of rutin and quercetin, studied in vivo and in vitro, leads to the formation of PA, mainly 3,4-dihydroxyphenylaceticacid (3,4-diOHPAA), 3-hydroxyphenylaceticacid (3-OHPAA), 4-hydroxybenzoicacid (4-HBA), 3,4-dihydroxybenzoicacid (3,4-diOHBA), and 3-hydroxyphenylpropionicacid (3-OHPPA) [1]. The recovery of quercetin in the ileal fluid of ileostomy patients was 83–86% [13,14] when ingested in its glycosidic form (rutin) and ~77% when ingested as quercetin aglycones [13,15].

The bioavailability of polyphenols is also dependent on their interaction with other substrates in the presence of the gut microbiota. Polyphenols are, most often, ingested alongside dietary fibres, sometimes in the same foods, and may be found together in plant cell walls. Therefore, they enter the colon at similar time.

Both fibre and polyphenols have been proposed to alter the growth and diversity of the gut microbiota, by which they are both catabolised. Most dietary fibres are fermented to SCFAs resulting in a fall in colonic pH. Some fibres selectively increase the populations of specific groups of bacteria, and therefore, act as prebiotics [16,17]. This, along with lower pH may alter the metabolism of some compounds. Both antibacterial and prebiotic effects have also been described for (poly)phenol compounds [18,19,20,21,22,23] and PA have also demonstrated antimicrobial impact especially towards pathogenic bacteria, with less impact on the commensal bacteria [24]. Aglycones such as quercetin have been shown to have stronger antibacterial properties than their glycosidic forms such as rutin.

Thus, polyphenols and fibres are converted to bioactive molecules by the gut bacteria and can also influence the composition of the microbiota. There is very little evidence, however, of any interactions between the polyphenols and fibre, and the resultant release of PA and SCFAs. Inclusion of a fermentable carbohydrate (glucose) in the fermentation media of faecal batch cultures, accelerated the degradation of rutin, releasing quercetin and further metabolites [1]. However, there is a lack of evidence on the interactions between dietary fibres, polyphenols, and the gut microbiota.

In this study, we investigated such interactions in vitro using a range of dietary fibres. Raftiline, ispaghula (psyllium), and pectin were chosen for their varying viscosity and fermentation rates as some of the potential inhibitory effect of fibre on polyphenol bioavailability could be linked to their viscosity and their fermentability [25]. Rutin and quercetin were chosen as they are very abundant in the diet alongside dietary fibre.

We aimed to determine whether the three fermentable fibres influenced the colonic microbiota metabolism of rutin and the release of PA, using an in vitro fermentation model. The impact of quercetin, rutin, and their colonic metabolites on the production of SCFAs from the fermentation of soluble fibres by the gut microbiota was also evaluated.

## 2. Materials and Methods

### 2.1. Participants and Methods

#### 2.1.1. Faecal Bacterial Donors

A 24 h batch culture model of human colonic fermentation was conducted as described previously by Edwards and Parrett [26], using faecal samples from 10 healthy participants. The participants included four men and six women, BMI (Body Mass Index) 22.5 ± 3.0 (range 18.4 to 27.4), and aged 23 ± 3.8 years (range 19 to 33 years) provided faecal samples for this study.

Participants were non-smokers, not pregnant, and had not consumed antibiotics or probiotics 3 months prior to the study. Each faecal donor followed a low (poly)phenol diet for 3 days prior to the study, avoiding fruits, vegetables, cocoa products, and (poly)phenol-rich beverages such as wine, tea, coffee, and fruit juice. On day 4, participants provided a faecal sample, which was processed within 2 h of collection. The study was approved by the College of Medicine, Veterinary and Life Sciences Ethical Committee (ref FM04509). All study participants provided informed consent, in accordance with the declaration of Helsinki.

#### 2.1.2. Batch Culture Model of Human Colonic Fermentation

Each faecal sample was homogenised separately in phosphate buffer (pH 7) to produce a 32% faecal slurry. The pre-reduced fermentation medium (44 mL) and the slurry (5 mL) were combined in 100 mL autoclaved fermentation bottles. The batch cultures were made by adding 1 g of raftiline, ispaghula or pectin with or without rutin or quercetin (final concentration of 28 µmol/L) to the slurry mixtures (50 mL). The fibres were obtained from commercial sources as extracts: Ispaghula husk (Whole ispaghula husk—myprotein.com, Cheshire, UK), pectin (pectin apple—250 grade, BDH laboratory, Poole, UK), and raftiline (commercial name for inulin by Orafti^®^ supplied by Siber Hegner Ltd., Zurich, Switzerland). As fermentation of the dietary fibre sources (raftiline, ispaghula, and pectin) may have released PA themselves, this was checked by also incubating faecal samples with each fibre alone. A control culture containing faecal sample only was included for each subject as a blank sample.

The contents of the incubation bottles (in duplicate) were fermented in a shaking water bath at 37 °C at 60 strokes/min for 24 h. Aliquots of the fermentation culture fluid (6 mL) were collected at 0, 2, 4, 6, and 24 h and stored immediately at −80 °C.

The pH of each culture was measured using a micro combination pH electrode suitable for measuring pH in small volumes (Shelfscientific–Lazer Research Laboratories, Inc., Los Angeles, CA, USA). The volume of gas produced during fermentation was measured using a 3-way tap and a 50-mL graduated syringe [27]. If the gas expelled exceeded 50mL, the valve was closed, and the syringe was emptied. The procedure was then repeated to measure the amount of gas remaining in the bottle.

#### 2.1.3. Phenolic Acid Quantification by Gas Chromatography Mass Spectrometry

Phenolic acids were extracted and derivatised using an adapted protocol from Combet, Lean [28]. Briefly internal standard 2,4,5-trimethoxycinnamic acid (30 µL) was added to fermentations samples (1 mL), prior to acidification with 1M HCl. Samples were cooled at 4 °C prior to liquid–liquid extraction with 1.5 mL ethyl acetate twice. The upper organic phases were combined and dried at 37 °C under a gentle flow of nitrogen. The derivatisation reagent, N,O-bistrifluoroacetamide (BSTFA, 50 µL) was added, vials were flushed with nitrogen and maintained at 80 °C for 4 h. Hexane (450 µL) was added to the derivatised samples before analysis.

A Trace gas chromatograph (GC), equipped with a split/splitless injector and an AI3000 autosampler, was interfaced to a dual stage quadrupole (DSQ) mass spectrometer (Thermo Fisher, Hemel Hampsted, UK). Samples (1 µL) were injected in a split mode with a 25:1 ratio. The inlet temperature was set at 220 °C and the oven was programmed at an initial temp of 45 °C increasing up to 300 °C (ramp: 45–160–200–250–300° C). A 1.2 mL/min flow rate was set for the carrier gas (helium). Hexane was used as pre-wash solution and acetonitrile as post-wash solution. All samples were extracted duplicates.

A standard calibration solution comprising 23 PA was extracted and analysed alongside urine and faecal fermentation samples. The PA present in the standard calibration solution were analysed individually to confirm retention time (tR) and mass spectra. Quantification was achieved using 2.5–15 µg/mL calibration curves of the PA Average standard area ratios were used, and a coefficient of variance was calculated (R^2^ > 0.98).

#### 2.1.4. SCFA Quantification by Gas Chromatography—FID (Flame Ionization Detector)

Measurement of the of SCFA production was measured using the Laurentin and Edwards [29] method. Briefly, immediately after sampling, aliquots of faecal sample fermentation fluid were added in a ratio of 2:1 to bijoux bottles containing 1ml NaOH and stored at −20 °C for SCFA measurement. The SCFAs were analysed on a TRACE 2000 gas chromatograph equipped with a flame ionisation detector (GC-FID;). The oven was programmed at an initial temperature of 80 °C for an initial 1 min, increasing 15 °C/min, thereafter increasing to 210 °C; using a Zebron ZB-Wax capillary column (15 m × 0.53 mm id × 1 μm-film thickness, Phenomenex, Cheshire, UK). The carrier gas (nitrogen) was set at a flow rate of 30mL/min. An internal standard (2-ethylbutyrate, 73.8 mmol/L) and orthophosphoric acid (0.1 mL) were added to 0.8 mL of faecal slurry aliquots. Extraction was carried out 3 times with diethyl ether (3 mL). Samples were centrifuged, and fractions pooled. Extractions were done in duplicate for each sample.

### 2.2. Statistical Analysis

A general linear model ANOVA was used to test (1) the effect of fibres (raftiline, ispaghula, and pectin) on PA production from rutin; and (2) the effect of polyphenols (rutin and quercetin) on SCFA production from the fibres. This statistical model allowed for the analysis of paired data with repeated measures. After assessing for normality, a paired *t*-test model or 1-way Wilcoxon test was carried out to analyse data at the 24 h point. This analysis was essential as the general linear model considers change over time, therefore not taking into consideration differences at specific time points. This is crucial as soluble fibres differ in their rate of fermentation, resulting in slowly fermented fibres (e.g., ispaghula) to demonstrate an impact only at 24 h. All statistical analyses were conducted using Minitab-15 (State College, PA, USA).

## 3. Results

### 3.1. Phenolic Acid Production from Bacterial Catabolism of Rutin

Rutin incubation alone resulted in the production of 7 PA (Table 1): 4-hydroxy benzoic acid (4-OHBA), phenyl acetic acid (PAA), 3-hydroxy phenyl acetic acid (3-OHPAA), 4-hydroxy phenyl acetic acid (4-OHPAA), 3,4-dihydroxy phenyl acetic acid (3,4-diOHPAA), 3-hydroxy phenyl propionic acid (3-OHPPA), and 4-hydroxy phenyl propionic acid (4-OHPPA).

There was a high person-to-person variation in the concentration of these PA, demonstrated by high standard deviations as seen for total PA produced from rutin at 24 h (Table 2). To estimate the bacterial metabolism efficiency and breakdown of rutin into PA, a percentage of the total sum of PA produced from rutin catabolism was calculated. The total sum of PA retrieved was 11.9 ± 6.0 µg/mL, accounting for 70% of the rutin added to fermentation vessels (17.0 µg/mL).

### 3.2. Impact of Soluble Fibres on Phenolic Acid Production

The impact of each fibre on the sum of PA is shown in Figure 1. Raftiline had a marked impact on total sum of PA production (*p* < 0.01, 85% reduction), followed by pectin (*p* < 0.01, 78% reduction) over 24 h fermentation. Ispaghula also inhibited the total sum of PA production (*p* < 0.05, 42% reduction), however this was only seen at the 24 h time point and not across time. Raftiline and pectin had a much higher impact on inhibiting total sum of PA produced compared to ispaghula (*p* = 0.005 and *p* = 0.01). No difference was found between raftiline and pectin (*p* = 0.8).

With the exception of PAA, which is produced from a range of other molecules, raftiline had the most impact (85% reduction, *p* < 0.01) on the production of each individual PA, followed by pectin (78% reduction, *p* < 0.01) and ispaghula (42% reduction, *p* < 0.05 at 24 h only). In fact, exploratory analysis demonstrated higher (1.6-fold, *p* = 0.7) 3-OHPPA production from rutin in the presence of ispaghula. While this difference was not statistically significant, it merited further investigation. The high concentrations of 3-OHPPA produced from both, resulted in an additive effect of 3-OHPPA produced from both ispaghula and rutin. The additive effect of 3-OHPPA produced from both ispaghula and rutin obscured the true impact of ispaghula on inhibition of 3-OHPPA produced from rutin. To determine the inhibitory impact of ispaghula on 3-OHPPA production, a theoretical value was calculated as the sum of 3-OHPPA, produced from (i) ispaghula-only fermentation and, (ii) rutin-only fermentation. The theoretical value was then compared to the concentration obtained from when ispaghula and rutin were incubated together using a 1-way Wilcoxon test. There was no difference between the two.

All incubation combinations demonstrated an increase in 3,4-diOHPAA concentration up to 6 h up to 3.1 ± 4.2 µg/mL and plateaued thereafter. There was no production of 3-OHPAA up to 6 h and an increase in concentration up to 1.2 ± 1.6 µg/mL thereafter (Figure 2).

We also analysed the impact of the polyphenols rutin and quercetin on the fermentation of the soluble fibres. There was not sufficient evidence for an impact from (poly)phenols on pH, gas, acetate, propionate, butyrate or total or SCFA (sum of all SCFAs), production as a result of fibre fermentation over time or at 24 h (Table 3).

## 4. Discussion

This study has shown clear interactions between the inclusion of fibres and polyphenols in cultures of gut microbiota, which resulted in a change in the release of bioactive catabolites. In vitro fermentation of the polyphenolic compound quercetin-3-O-rutinoside led to the production of seven PA, as previously described [1,30,31,32,33]. The most abundant PA was PAA, followed by 3,4-diOHPAA, 3-OHPAA, 4-OHPAA, 3-OHPPA, 4-OHPPA, and 4-OHBA. Phenolic acid from the background diet, determined based on their presence in the faecal sample alone incubations, included PAA, 3-OHPAA (in one participant only), 4-OHPAA, 3-OHPPA, and 4-OHPPA. The differences detected for rutin metabolism between participants were transient production of 4-OHBA in 2/10 participants and 3-OHPAA in 9/10 participants (Table 2). It has been proposed that there are different metabotypes where individual microbiomes produce different patterns of metabolites, including PA from the same polyphenol source [34]. We did not detect any other difference in metabotypes from rutin metabolism between the participants (Appendix A). A larger study may be required to further investigate differences in metabotypes. The paired study model allowed the use of FS-alone incubation as a control, despite the presence of PA from the background diet.

The dietary fibres raftiline, ispaghula, and pectin were chosen due to their varying viscosity and rates and patterns of fermentation. We did not measure viscosity in our study. However, it was clear on visual assessment of the fermentation vessels that raftline was not viscous and remained liquid throughout the fermentation. Pectin and ispaghula were viscous, with ispaghula displaying higher viscosity and forming a thick gel. Pectin and Ispaghula maintained their viscosity throughout the fermentation.

Raftiline as a fibre with higher fermentability (raftiline) had greater inhibitory impact on the PA production from rutin catabolism than ispaghula, which had the highest viscosity and lower fermentability. We did not detect an impact of rutin or quercetin at a concentration of 28 µmol/L on the pH or SCFA and gas production from these fibres. These results are in line with an in vitro fermentation study by Aura, O’leary [31] which also demonstrated no impact from rutin and its metabolites on the pH of fermentation vessels.

The production of 3,4-diOHPAA in the current study and all previous studies demonstrates it as a dominant PA from rutin/quercetin colonic degradation, followed by 3-OHPAA. Previous studies did not detect PAA in high concentrations as was found in faecal incubations of this study. This may be attributed to LCMS analysis in previous studies compared to GCMS analysis used in this study. Phenyl acetic acid is non-polar in its non-conjugated form and has a low molecular weight [35]; therefore, it is better detected by the GCMS due to poor ionisation in the LCMS. An improved sensitivity of GCMS for the detection of PAA and PBA compared to low sensitivity of LCMS has been demonstrated in other studies [36,37,38]. Furthermore, FS-alone incubations resulted in 4.3 ± 1.6 µg/mL compared to 5.5 ± 3.3 µg/mL of PAA production from rutin incubations. Thus, the majority (77.4%) of PAA detected in our study was from the background diet.

Contrary to previous reports by Jaganath, Mullen [1] and Aura, O’leary [31], we did not detect 3,4-DiOHBA or 3,4-DiOHPPA. This may be due to rapid conversion of 3,4-DiOHPPA to 4-OHPPA and 3,4-DiOHPPA to 3-OHPPA, which were both identified in our samples. Our findings support the findings of Aura et al. [31], that 3-OHPAA increases at 6 h concomitantly with the plateauing of 3,4-diOHPAA production. This suggests that 3-OHPAA is produced as a result of further degradation of 3,4-diOHPAA (Figure 2).

In this study, 3,4-diOHPAA, PAA and total PA were inhibited by raftiline (81.6%, 92.5% and 85.5% reduction, respectively), pectin (66.8%, 95%, 78.1% reduction, respectively) and ispaghula (27.6%, 47.2%, 42.3% reduction, respectively). While 3-OHPAA and 4-OHPAA were inhibited by only raftiline (98.4%, 76% reduction, respectively) and pectin (89.6%, 74.6% reduction, respectively); 3-OHPPA and 4-OHPPA were not inhibited by any of the fibres. Only 2/10 participants produced 4-OHBA in rutin incubation and 1/10 in raftiline + rutin incubations. Pectin and raftiline completely inhibited the production of 4-OHBA (Table 1).

The highly fermentable fibres such as raftiline exhibited an inhibition of PA production as early as 2 h post fermentation; ispaghula having a slower fermentation rate demonstrated an impact only at the 24 h sampling point and not across time, with the exception of its impact on PAA which was seen across time.

Raftiline and pectin had a higher inhibition percentage on total PA production than ispaghula. No difference was seen between raftiline (85.5% reduction) and pectin (78.1% reduction). Raftiline also inhibited PA production from polyphenolics present in the background diet, which was shown in higher PA concentration in FS-only incubations compared to raftiline only incubations.

These results are novel and demonstrate the importance of considering the impact of fibre viscosity and fermentability when studying their impact on the gut microbiota and their influence on PA production. There is insufficient evidence on the impact of carbohydrates on colonic metabolism of rutin.

In this study, the impact of fermentable fibres with low and high viscosity and fermentation rates on the degradation of rutin by colonic bacteria were considered. This enabled better understanding of the role of viscosity or fermentability on the bioavailability of polyphenol metabolite.

Most studies have been designed to investigate the antibacterial properties of polyphenols [18,39,40,41,42] and not their impact on the colonic metabolite production of the microbiota. Such studies can only provide an assumption that the observed antibacterial activity of polyphenols will alter colonic metabolite production, such as SCFA production from fibre. Our study did not show any impact of polyphenolics on SCFA production from fibre catabolism. Additionally, PA production had no effect on incubation media pH or gas production.

Dietary fibres and dietary polyphenols may compete for catabolism by gut bacteria and their catabolism may affect the other [9]. The consideration of the prebiotic potential of fibres in conjunction to the antibacterial properties of the polyphenols is of importance. The potential impact of rutin and quercetin on the microbiota may have been ameliorated by the prebiotic properties of raftiline, pectin, and ispaghula. The findings of this study are limited by the absence of microbial assessment. Focusing on microbial assessment in association with their catabolic by-products and measurement of fibre viscosity in future studies would provide further insight to the impact of fibres and polyphenols on the microbiota.

Most studies [20,39,41] investigating the effects of polyphenols on gut microbiota have used high concentrations of polyphenols over long periods of time predominantly for their potential application in the food and pharmacology industry. They have not taken into consideration the bioavailability of these polyphenols. The concentration of rutin and quercetin (17.0 µg/mL rutin and 8.46 µg/mL quercetin) used in this study are based on their physiological bioavailability as recovered in the ileal fluid of patients 0–24 h after consuming tomato juice supplemented with 175 µmol/L rutin [43]. This was much lower than the lowest MIC found for these compounds in vitro. We did not measure the changes in bacterial populations in the study.

Some PA were found in cultures of faecal samples alone. The reason for this is unclear. Compliance to the low polyphenol diet was shown by the dietary record of participants analysed using the Phenol Explorer database [44].

It is important to note that our findings did not confirm whether the impact of fibres on PA production was inhibitory or just retardation. The total sum of PA showed a linear increase suggesting that the final total amount could have been much higher if more time were available. As demonstrated by Pérez-Jiménez, Serrano [45], the release of antioxidants associated with dietary fibre was delayed.

The role of the microbiota in health and disease is currently a subject of considerable interest [46,47]. Previous studies reporting health benefits from consumption of polyphenols or fibre did not take into consideration the interaction of these two components, which are most often found in combination in foods. The health claims for fibre has led the food industry to incorporate fibre into many products which are naturally high in polyphenols. This may ameliorate the potential health benefits of polyphenols, and is therefore of relevance to the food industry. This study shows that the impact of fibres on bioavailability of food polyphenols and subsequent benefits on health needs to be considered in this regard, with careful evaluation of all parameters that may influence health outcomes, from the perspective of the individual, the food chosen, and the study methods.

## 5. Conclusions

Highly fermented fibres inhibited PA production from the catabolism of rutin by human faecal bacteria, whereas rutin and quercetin had no impact on SCFA production from the fermentation of these fibres in vitro. When considering the health benefits and the bioavailability of polyphenols, the impact of fibre on their metabolism in the colon should be considered.

## Figures and Tables

**Figure 1 ijerph-16-00292-f001:**
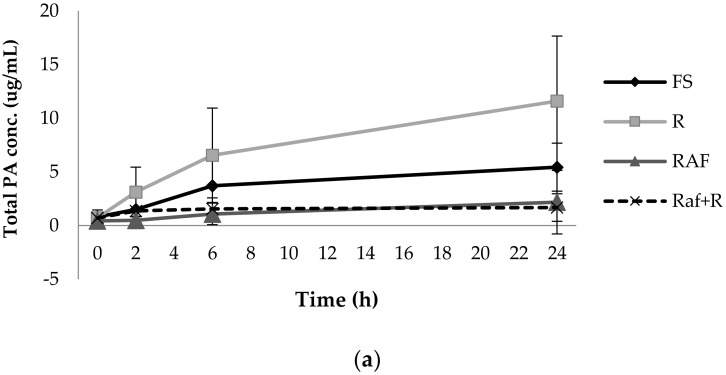
Impact of raftiline (**a**), ispaghula (**b**), and pectin (**c**) on total phenolic acid (PA) production from rutin incubation with human faecal bacteria. The inhibitory impact of raftiline, ispaghula, and pectin across time are demonstrated as mean (± SD). FS: Faecal slurry, R: Rutin, RAF: Raftiline, RR: Raftiline + Rutin, ISP: Ispaghulla, ISP+R: Ispaghula + Rutin, Pec: Pectin, PEC+R: Pectin + Rutin. Rutin and FS only incubations were matched for all groups.

**Figure 2 ijerph-16-00292-f002:**
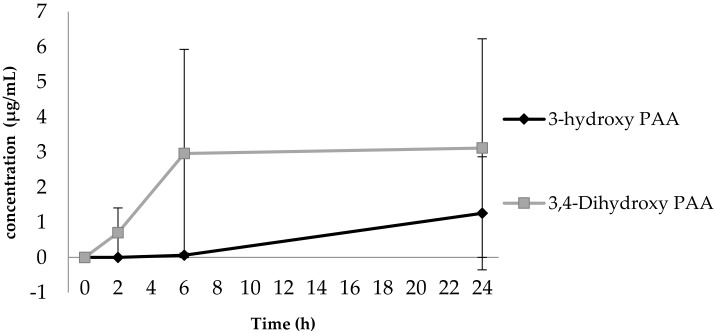
Production of 3,4-diOHPAA from the degradation of 3-OHPAA. The plateauing of 3,4-DiOHPAA at 6 h concomitantly to the increase in production of 3-OHPAA at 6 h is demonstrated as mean (± SD).

**Table 1 ijerph-16-00292-t001:** Phenolic acids identified in fermentations containing rutin only.

Peak	Retention Time (*t_R_*) in Minutes	Identifying Ion (m/z)	Qualifying Ions (m/z)	Phenolic Acid
Peak 1	7.45	164	73, 91, 164	4-OHBA
Peak 2	13.22	164	73, 147, 75	PAA
Peak 3	13.82	267	73, 267, 193	3-OHPAA
Peak 4	14.15	179	73, 164, 281, 252	4-OHPAA
Peak 5	17.17	205	192, 177, 73, 310	3,4-diOHPAA
Peak 6	18.44	179	192, 73, 75, 177	3-OHPPA
Peak 7	21.1	179	73, 267, 384	4-OHPPA

Retention time, identifying ion, and qualifying ions were used to identify the polyphenolics in this study.

**Table 2 ijerph-16-00292-t002:** PA (phenolic acids) concentrations in fermentations after 24 h.

	4-OHBA (µg/mL)	PAA (µg/mL)	3-OHPAA (µg/mL)	4-OHPAA (µg/mL)	3,4-diOHPAA (µg/mL)	3-OHPPA (µg/mL)	4-OHPPA (µg/mL)
Detection frequency	2/10	10/10	9/10	10/10	10/10	10/10	10/10
FS	-	4.33 ± 1.67	0.006 ± 0.02	0.50 ± 0.43	-	0.21 ± 0.14	0.75 ± 1.23
R	0.04 ± 0.02	5.59 ± 3.35	1.25 ± 1.60	0.79 ± 0.66	3.11 ± 4.27	0.43 ± 0.28	0.39 ± 0.68
RAF	-	0.72 ± 1.29	0.002 ± 0.007	0.18 ± 0.24	-	0.24 ± 0.18	0.15 ± 0.16
RR	0.05 ± 0.0	0.41 ± 0.90	0.02 ± 0.06	0.19 ± 0.25	0.56 ± 0.92	0.34 ± 0.15	0.12 ± 0.10
% Change	↑40%	↓92.5% **	↓98.4% **	↓76% **	↓81.6% **	↓20.9%	↓39.1%
ISP	-	2.19 ± 1.86	0.002 ± 0.007	0.25 ± 0.46	-	0.39 ± 0.43	0.56 ± 0.71
ISP-R	-	2.94 ± 2.61	0.96 ± 1.24	0.33 ± 0.62	2.29 ± 2.86	0.69 ± 0.80	0.47 ± 0.69
% Change	↓100% **	↓47.2% **	↓23.2%	↓58.2%	↓27.6% *	↑37.6%	↑10.8%
PEC	-	0.28 ± 0.57	0.001 ± 0.005	0.16 ± 0.18	-	0.25 ± 0.13	0.08 ± 0.09
PR	-	0.29 ± 0.54	0.12 ± 0.26	0.2 ± 0.21	1.03 ± 1.60	0.39 ± 0.18	0.1 ± 0.1
% Change	↓100% **	↓95% **	↓89.6% **	↓74.6% **	↓66.8% **	↓9.30%	↓74.3% **

Results are shown as mean values (± SD) at 24 h in 50 mL faecal incubations having 28 µmol/L rutin with 1 g of fibre (n = 10). * *p* < 0.05 ** *p* < 0.01, FS: Faecal slurry, R: Rutin, RAF: Raftiline, RR: Raftiline + Rutin, ISP: Ispaghulla, ISP+R: Ispaghula + Rutin, Pec: Pectin, PEC+R: Pectin + Rutin, ↑ increase, ↓ decrease. Detection frequency refers to number of participants out of total participants producing the PA. % Change is calculated as percentage inhibition of the PA production when comparing fibre-rutin combination to rutin only incubation.

**Table 3 ijerph-16-00292-t003:** Total short-chain fatty acid (SCFA) production (6 and 24 h post-fermentation), gas, and pH measurement (at 24 h).

Substrate	Total SCFA Concentration at 6 h (mmol/L)	Total SCFA Concentration at 24 h (mmol/L)	pH at 24 h	Gas Production at 24 h (mL)
Raftiline	49.6 ± 8.4	80.3 ± 8.2	4.27 ± 0.4	21.0 ± 10.4
Raftiline + Rutin	42.6 ± 11.3	76.6 ± 7	4.68 ± 0.88	23.2 ± 11.9
Raftiline + Quercetin	41.9 ± 17.8	80.6 ± 7.4	4.30 ± 0.50	26.9 ± 8.0
Ispaghula	28.5 ± 11.8	76.8 ± 7.8	5.33 ± 0.53	20.7 ± 4.1
Ispaghula + Rutin	29.1 ± 9.5	66.2 ± 4.5	5.46 ± 0.64	17.5± 4.2
Ispaghula + Quercetin	28.6 ± 9.7	63 ± 16.5	5.42 ± 0.63	15.7 ± 6.4
Pectin	39.9 ± 15.3	68.7 ± 47.6	3.92 ± 0.64	22.7 ± 12.4
Pectin + Rutin	38.8 ± 9.3	67.5 ± 21.8	4.11 ± 0.60	21.6 ± 10.4
Pectin + Quercetin	38 ± 13.6	59.8 ± 29.2	4.01 ± 0.55	17.7 ± 8.2

Values are mean (± SD) at 6 and 24 h for 50 mL faecal incubations having 28 µmol/L rutin/quercetin with 1 g of fibre.

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
