# Peer review of "Impact of Fermentable Fibres on the Colonic Microbiota Metabolism of Dietary Polyphenols Rutin and Quercetin"

_ijerph, 2019, doi:10.3390/ijerph16020292_

Round 1

Reviewer 1 Report

This manuscript details a study of phenolic acid (PA) production by human fecal bacteria with and without the addition of either polyphenols (which can be digested into PA) or fermentable fibers (which inhibit PA production). The results are presented well and I believe the strength of this paper is showing the potentially antagonistic relationship between these two classes of compounds commonly used as prebiotics. While rutin appears to have no effect on the digestion of fiber into short chain fatty acids, all three fibers inhibit production of PA (whether from rutin or from other sources). It is unclear to me from this data whether there is any special effect of these fibers on the metabolism of rutin and quercetin as opposed to the general effect on PA production. I have two suggestions for the authors, detailed below; 

1) Because the processes used to obtain fermentable fibers vary considerably, the authors should provide some further description of how exactly the samples used in this study were obtained.

2) I believe this work could be significantly strengthened by basic examination of the microbes in these fecal samples. Even simple 16S sequencing of the 10 samples could be helpful in explaining the variability in PA production and if this could be combined with some longitudinal study of the change in microbiome over the 24 fermentation period, we might learn a great deal about the mechanism of inhibition by the fibers. 

Author Response

Thank you very much for your comments. We have edited the manuscript accordingly and responded to the comments below.

1) Because the processes used to obtain fermentable fibers vary considerably, the authors should provide some further description of how exactly the samples used in this study were obtained.

Response: Thank you, the information was added at line 112 of manuscript: the fibres were obtained from commercial sources as extracts. Ispaghula husk (Whole ispaghula husk – myprotein.com), pectin (pectin apple- 250 grade, BDH laboratory supplies), and raftiline (Orafti® supplied by Siber Hegner Ltd).

2) I believe this work could be significantly strengthened by basic examination of the microbes in these fecal samples. Even simple 16S sequencing of the 10 samples could be helpful in explaining the variability in PA production and if this could be combined with some longitudinal study of the change in microbiome over the 24 fermentation period, we might learn a great deal about the mechanism of inhibition by the fibers.

Response:  We agree that some microbiome characterization would have been useful; however the DNA was not extracted from these samples in an appropriate timeframe (controlling and uniformisation of time periods from sample aliquoting, storage, extraction, storage, analysis) making such analysis unfeasible and possibly misleading.  Detailed examination of the association of the microbiome with PA production needs a bigger study focused on this aspect.  We have discussed this limitation in the manuscript.

Reviewer 2 Report

This study explores the effect of co-fermentation of selected polyphenols and fermentable dietary fibre  on microbial metabolism of polyphenols (monitored by measuring the phenolic acids) and fibre (SCFA). The study lies at the interface between food science and nutrition. Its originality lies in the fact that interactions between PP and DF on the reciprocal metabolism are explored which has been poorly considered so far. Despite the limitedness of the experimental plan and the obvious limitation of a batch fermentation model, I suggest the publication of the study after the following issues have been addressed:

Line 37-39 and line 44-45: there is no need to report all the details of those two studies

Page 2, line 53: The products of bacterial fermentation in the colon (at least not all) are not specific, metabolism of several other PP may produce the same compounds

Is it raftiline a commercial name? shouldn’t the authors indicate that and possibly indicate its nature?

Line 165-166: Move that info in Materials and Methods

Line 175 compared to line 177: is it 11 microg/L or 11.9 microg/L? specify

The authors may discuss their findings in the light of the existence of different metabotypes for PP metabolism. Was the difference in rutin metabolism also apparent in terms of type of PA produced? For instance, were there subjects only producing some PA compared to others? This is partly reported in table 1 but further information might be extracted from reporting the concentration data for each single donor (this may also be done by providing the table as supplementary material)

The authors suggest that the effect of fibre on PP metabolism may be due to difference in viscosity. Did they try to measure that viscosity? Is the difference that large to justify the observed difference or is the difference kept along fermentation? I suggest they either measure that viscosity or provide literature data on the differences in viscosity that are expected from the fibre concentrations they have used.

Conclusions may be enriched with considerations on the wider impact of this research or next moves in the field. At the moment it looks like a mini-abstract of the study.

Author Response

Thank you very much for your comments. We have edited the manuscript accordingly and responded to the comments below.

1) Line 37-39 and line 44-45: there is no need to report all the details of those two studies

Response:  Thank you for the comments.  We have deleted these details.

2) Page 2, line 53: The products of bacterial fermentation in the colon (at least not all) are not specific, metabolism of several other PP may produce the same compounds

Response: This has been corrected in the manuscript: Colonic metabolism of rutin and quercetin, studied in vivo and in vitro leads to the formation of PA, mainly 3,4- dihydroxyphenylaceticacid (3,4-diOHPAA), 3-hydroxyphenylaceticacid (3-OHPAA), 4-hydroxybenzoicacid (4-HBA), 3,4-dihydroxybenzoicacid (3,4-diOHBA), and 3- hydroxyphenylpropionicacid (3-OHPPA) [1].

3) Is it raftiline a commercial name? shouldn’t the authors indicate that and possibly indicate its nature?

Response:  Thanks for your comment.  We have now explained the nature of the raftline used in the manuscript at line 114: Raftiline (trade name for inulin by Orafti® supplied by Siber Hegner Ltd ) was used.

4) Line 165-166: Move that info in Materials and Methods

Response: This has been moved.

5) Line 175 compared to line 177: is it 11 microg/L or 11.9 microg/L? specify

Response: The correct value is 11.9 microg/L and this has now been corrected.

6) The authors may discuss their findings in the light of the existence of different metabotypes for PP metabolism. Was the difference in rutin metabolism also apparent in terms of type of PA produced? For instance, were there subjects only producing some PA compared to others? This is partly reported in table 1 but further information might be extracted from reporting the concentration data for each single donor (this may also be done by providing the table as supplementary material)

Response: There was no difference in type of PA produced from rutin metabolism between participants. Differences detected were transient production of 4-OHBA in 2/10 participants and 3-OHPAA in 9/10 participants. The only other difference detected was in concentration of PA produced for each participant. We have now discussed this in the manuscript at line 236. A table has been prepared to be attached as a supplementary table.

Added to manuscript: The differences detected for rutin metabolism between participants were transient production of 4-OHBA in 2/10 participants and 3-OHPAA in 9/10 participants. It has been proposed that there are different metabotypes where individual microbiomes produce different patterns of metabolites including PAs from the same polyphenol source [36]. We did not detect any other difference in metabotypes from rutin metabolism between the participants. A larger study may be required to further investigate differences in metabotypes.

Supplementary table-1: PA production during rutin incubation in in vitro fecal batch cultures for each participant

V   -3

V-4

V -5

V -6

V -7

V -8

V -9

V -10

V -11

V -12

4-OHBA (µg/ml)

R-0 h

0.00

0.00

R-2 h

0.06

0.00

R-6 h

0.06

0.00

R-24 h

0.05

0.02

PAA        (µg/ml)

R-0 h

0.57

0.15

0.35

0.15

0.40

0.79

0.29

0.23

0.42

0.00

R-2 h

0.67

0.73

0.87

0.58

0.38

0.57

0.43

1.22

1.90

2.63

R-6 h

1.07

0.47

1.56

0.42

0.64

1.03

1.56

5.66

5.08

1.66

R-24 h

4.64

2.37

11.62

3.59

1.25

4.16

6.31

4.94

6.27

10.83

3-OHPAA (µg/ml)

R-0 h

0.00

0.00

0.00

0.00

0.00

0.00

0.00

0.00

0.00

0.00

R-2 h

0.00

0.00

0.00

0.00

0.00

0.00

0.00

0.00

0.00

0.00

R-6 h

0.00

0.00

0.10

0.20

0.04

0.00

0.00

0.09

0.06

0.09

R-24 h

0.06

0.08

0.52

2.40

0.07

0.00

1.89

0.54

1.97

5.03

4-OHPAA (µg/ml)

R-0 h

0.08

0.04

0.00

0.01

0.07

0.28

0.04

0.08

0.12

0.84

R-2 h

0.21

0.18

0.15

0.14

0.22

0.21

0.19

1.97

2.13

2.41

R-6 h

0.50

0.08

0.23

0.12

0.25

0.74

0.20

0.69

1.79

1.72

R-24 h

0.49

2.18

0.65

0.22

0.55

1.18

0.10

0.11

1.30

1.16

3,4- diOHPAA (µg/ml)

R-0 h

0.00

0.00

0.00

0.00

0.00

0.00

0.00

0.00

0.00

0.00

R-2 h

0.44

0.14

0.19

0.24

0.36

0.32

1.69

1.15

1.13

1.37

R-6 h

1.68

0.01

1.53

3.15

0.82

4.15

4.75

2.51

2.11

8.90

R-24 h

5.27

0.46

7.31

0.32

5.24

12.36

0.02

0.08

0.02

0.03

3-OHPPA (µg/ml)

R-0 h

0.10

0.02

0.00

0.31

0.16

0.38

0.30

0.06

0.27

0.59

R-2 h

0.18

0.22

0.26

0.70

0.32

0.27

0.42

0.20

0.52

0.37

R-6 h

0.23

0.03

0.23

0.28

0.21

0.50

0.67

1.08

1.54

0.72

R-24 h

0.23

0.05

0.77

0.40

0.56

0.79

0.37

0.10

0.27

0.78

4-OHPPA (µg/ml)

R-0 h

0.00

0.00

0.00

0.06

0.00

0.15

0.00

0.00

0.00

0.00

R-2 h

0.08

0.00

0.18

0.30

0.18

0.18

0.20

0.24

0.80

0.76

R-6 h

0.00

0.00

0.43

0.07

0.09

0.27

1.01

0.24

1.88

0.27

R-24 h

0.00

0.33

0.09

0.13

0.58

2.27

0.06

0.02

0.04

0.37

Values are mean of duplicate measurements at 0, 2, 6 and 24 h for rutin only incubations. R: rutin, h: hour

7) The authors suggest that the effect of fibre on PP metabolism may be due to difference in viscosity. Did they try to measure that viscosity? Is the difference that large to justify the observed difference or is the difference kept along fermentation? I suggest they either measure that viscosity or provide literature data on the differences in viscosity that are expected from the fibre concentrations they have used.

Response: We did not measure viscosity in our study. However, visual assessment of the fermentation vessels demonstrated that raftline was not viscous and remained as a liquid throughout the fermentation. Pectin and ispaghula were viscous, with ispaghula displaying higher viscosity and forming a thick gel. Pectin and Ispaghula maintained their viscosity throughout the fermentation.  We have not referred to literature values as viscosity can be impacted by parameters such dilution in gut contents, sheer rate and acidification. However, it is well established that ispaghula and pectin are viscous polysaccharides and inulin is not.  We have now discussed this in the manuscript. 

8) Conclusions may be enriched with considerations on the wider impact of this research or next moves in the field. At the moment it looks like a mini-abstract of the study.

Response: Thank you. We have made the following changes in the manuscript:

The role of the microbiota in health and disease is currently a subject of considerable interest [47, 48].

Previous studies reporting health benefits from consumption of polyphenols or fibre did not take into consideration the interaction of these two components, which are most often found in combination in foods. The health claims for fibre has led the food industry to incorporate fibre in many products which are naturally high in polyphenols. This may ameliorate the potential health benefits of polyphenols and is therefore of relevance to the food industry. This study shows that the impact of fibres on bioavailability of food polyphenols and subsequent benefits on health needs to be considered in this regard, with careful evaluation of all parameters that may influence health outcomes, from the perspective of the individual, the food chosen and the study methods.

Reviewer 3 Report

There is a growing interest in the role of bacterial metabolites in health and disease. This study looks at the interaction of metabolites of fiber (SCFA) and polyphenols (phenolic acids (PA)) in 10 faecal batch cultures from healthy humans after a low (poly)phenol diet 3 days prior to study. Production of all PA except two was reduced by at least one fiber. This effect was more pronounced for rutin, then pectin and ispaghula. No phenol effect on SCFA production was observed.

The findings of this study adds to the current knowledge with regard to interaction of the SCFA and PA. Data presentation and discussion could be improved to improve the quality of the manuscript as suggested below.

-Overall the manuscript is lengthier than it should be, with large amount of dispersed information, especially in the introduction and discussion, which if cut and focused, could read better. I suggest halving the introduction and making it more focused, leading to the hypothesis of the current study and its aim.

Discussion is dispersed with scattered facts of the current study and those from others. Instead discussion should aim to interpret the results of this study within the frame of the existing evidence. For example in lines 282-285, a study was quoted on the inhibitory effect of fiber and role of fiber viscosity on polyphenol bioavailability and that the study compared reported on fermentable fibers with low and high viscosity on the degradation of rutin by colonic bacteria. How these findings are related to the results of the current study? This we do not see. Discussion should report highlights of the current study, and interpret them within the context of previous findings such as this one.

Other points:

- On the effect of SCFA on PA: Authors conclude here that these findings could suggest no effect of fiber diet on polyphenol diet. However fiber, other than production of SCFA, could result in microbiota changes in vivo which could not be assessed by the assay used in this current study. So the conclusion should be softened here.

-Did authors differentiate effects of Rutin on each individual SCFA (butyrate, acetate, propionate)? If so would be interesting to know.

In lines 272-273: Authors report that “These results are novel and demonstrate the importance of considering the impact of fibre viscosity and fermentability when studying their impact on the gut microbiota and their influence  on PA production.”. Gut microbiota was not looked at in this study. Limitations of the study, such as absence of microbiota assessment, or absence of careful dietary evaluation (and their correlation with the findings of the study) should be endorsed and listed in the discussion by the authors.

-In line 290, authors report that their results “did not show any impact of polyphenolics on the activity of the microbiota on fibres and subsequent SCFA production”. However they cannot extrapolate their findings to the gut microbiota as above.

-They stated that “The inhibition of PA production in the presence of fibres, and lack of effect of PA on SCFA production from fibres in our study may indicate as expected that fibre is the preferred source of energy for the gut bacteria.” Can they elaborate on that and how this conclusion is made?

Author Response

Thank you very much for your comments. We have edited the manuscript accordingly and responded to the comments below.

1) Overall the manuscript is lengthier than it should be, with large amount of dispersed information, especially in the introduction and discussion, which if cut and focused, could read better. I suggest halving the introduction and making it more focused, leading to the hypothesis of the current study and its aim.

Response: Thank you for your comments, we have edited the introduction  by making it more focused and removing details of other studies which may not be directly leading to our study hypothesis.

2) Discussion is dispersed with scattered facts of the current study and those from others. Instead discussion should aim to interpret the results of this study within the frame of the existing evidence. For example in lines 282-285, a study was quoted on the inhibitory effect of fiber and role of fiber viscosity on polyphenol bioavailability and that the study compared reported on fermentable fibers with low and high viscosity on the degradation of rutin by colonic bacteria. How these findings are related to the results of the current study? This we do not see. Discussion should report highlights of the current study, and interpret them within the context of previous findings such as this one.

Response: Thank you, we have edited the discussion. We have shortened the discussion by removing details of studies that are not directly reflecting of our study results. We have also focused on study results and expanded on some of these results. We have also expanded on the limitations of the study.

3) On the effect of SCFA on PA: Authors conclude here that these findings could suggest no effect of fiber diet on polyphenol diet. However fiber, other than production of SCFA, could result in microbiota changes in vivo which could not be assessed by the assay used in this current study. So the conclusion should be softened here.

Response: Thank you, this has been corrected in the manuscript:  In the conclusion (line 388) It has been added that the lack of impact of PP on SCFA is in-vitro.

4) Did authors differentiate effects of Rutin on each individual SCFA (butyrate, acetate, propionate)? If so would be interesting to know.

Response: The Impact of rutin and quercetin was determined for acetate, propionate and butyrate individually. The sum of these 3 SCFAs and all other SCFAs made up the Total SCFAs. There was not sufficient evidence for an impact from rutin and quercetin on acetate, propionate, butyrate or Total SCFAs. This has been added to the manuscript at Line 243. 

5) In lines 272-273: Authors report that “These results are novel and demonstrate the importance of considering the impact of fibre viscosity and fermentability when studying their impact on the gut microbiota and their influence  on PA production.”. Gut microbiota was not looked at in this study. Limitations of the study, such as absence of microbiota assessment, or absence of careful dietary evaluation (and their correlation with the findings of the study) should be endorsed and listed in the discussion by the authors.

Response: Thank you, we have now discussed these limitations at line 331:  The findings of this study are limited by the absence of microbial assessment. Focusing on microbial assessment in association with their catabolic by-products and measurement of fibre viscosity in future studies would provide further insight to the impact of fibres and polyphenols on the microbiota.

6) In line 290, authors report that their results “did not show any impact of polyphenolics on the activity of the microbiota on fibres and subsequent SCFA production”. However they cannot extrapolate their findings to the gut microbiota as above.

Response: This has been corrected in the manuscript.

7) They stated that “The inhibition of PA production in the presence of fibres, and lack of effect of PA on SCFA production from fibres in our study may indicate as expected that fibre is the preferred source of energy for the gut bacteria.” Can they elaborate on that and how this conclusion is made?

Response: The original statement has been removed from the manuscript, The statement related to the role of dietary fibres as the main energy source in the colon and  is also based on the results of our study; where highly fermentable fibres had a higher inhibitory impact on the polyphenols compared to less fermentable fibres along with the lack of impact of polyphenols on the fermentation of fibres.  However, we have not demonstrated the preferred source of energy directly so have removed the sentence.
